# The Role of Place Attachment in Promoting Refugees’ Well-Being and Resettlement: A Literature Review

**DOI:** 10.3390/ijerph182111021

**Published:** 2021-10-20

**Authors:** Thomas Albers, Silvia Ariccio, Laura A. Weiss, Federica Dessi, Marino Bonaiuto

**Affiliations:** 1Department of Psychology of Developmental and Socialization Processes, Sapienza University of Rome, 00185 Rome, Italy; silvia.ariccio@uniroma1.it (S.A.); federica.dessi@uniroma1.it (F.D.); marino.bonaiuto@uniroma1.it (M.B.); 2Faculty of Social and Behavioural Sciences, Department of Social, Health and Organisational Psychology, Utrecht University, 3584 CS Utrecht, The Netherlands; l.a.weiss@uu.nl; 3Optentia Research Focus Area, North-West University, Vanderbijlpark 1900, South Africa; 4CIRPA–Interuniversity Research Centre of Environmental Psychology, Sapienza University of Rome, 00185 Rome, Italy

**Keywords:** well-being, refugees, place attachment, self-determination theory, integration

## Abstract

Refugees are at great risk of developing mental health problems. Yet, little is known about how to optimally help this vulnerable group as there is a lack of evaluated refugee mental health interventions. The current article presents the results of a literature review which investigates the importance of place attachment for the promotion of refugees’ well-being in the resettlement process. This review concentrated on the most recent and current literature regarding the potential role, importance, and relevance of people–place bonds in the dynamic process of refugee resettlement. It examines literature from the field of positive and environmental psychology, highlighting key theoretical concepts and research findings as well as gaps in research. The review revealed that little is known about the dynamics of place bonding, while the debate rages on about the geometry of the psychological constructs of person–place relationships. Yet, knowing more about which needs should be satisfied for easing place bonding could be of crucial importance for facilitating refugee well-being. Ultimately, improving the knowledge and understanding of the phases of this dynamic process could be useful for a more successful implementation of refugee resettlement practices and activities.

## 1. Introduction

Good health and well-being, decent work and economic growth, and reduced inequalities are three of the UN sustainable development goals for 2030 [1]. Although many efforts have been made, this goal is still out of reach, especially for refugees. It is therefore relevant to examine the current protocols for and future perspectives on promoting refugees’ well-being and positive integration into society. We will give a summary of the present knowledge and relevant literature and the potential role of people–place bonding on the promotion of refugee well-being. Implications for practice and future research are described in order to foster knowledge and understanding of this global issue.

The available literature on the role of people–place bonds in the promotion of refugees’ well-being is scarce, fragmented, and incomplete. This paper aims to fill this gap by identifying and examining the most recent and relevant literature which deals with the role refugees’ place-bonding dynamics have for their wellbeing and how these bonds develop over time in new settlements. The following question guided this review: “What is the potential relevance of people–place bonds for the promotion of refugee well-being?”.

## 2. Method

A literature review synthesizes published material from diverse sources covering a wide range of subjects [2]. Grant and Booth [2] note that a literature review usually includes material with a degree of permanence and possibly has been peer-reviewed. This type of review takes stock of the current body of work and provides a synthesized textual analysis of its contribution and value.

For the current study, the databases of PsycINFO, Web of Science and Scopus were used to search for literature. Papers included in this review were published in international peer-reviewed journals. The search terms used were *refugees/migrants/immigrants/migration,* combined with *well-being/positive psychology interventions, positive mental health, place attachment**/place identity place meaning/sense of place/place bonding/place making/self-determination theory/continuity* or *integration*. The search terms and criteria were deliberately kept broad to cast the net as wide as possible since there is a limited body of research available on the topic. For similar reasons, no restriction on publication date was imposed. As the literature on refugees is sparse, papers related to migrants were also included in this review. Finally, 79 papers were included in the paper (studies used for the review on refugees and migrants are marked with an asterisk (*) in the reference list), of which 17 specifically related to refugees and 15 to migrants.

The literature from, e.g., positive psychology, environmental psychology, etc., was synthesized and is presented as a comprehensive narrative. Gaps in research are identified, and the relevance of the results for the practice of the promotion of refugees’ well-being is discussed.

## 3. Results

The results of the review are presented in three sections. First, insight gained from the reviewed literature about refugee mental ill-health, mental well-being, and the relevance of mental health well-being interventions is presented. Second, Self-Determination Theory as a strategy/framework for positive integration is reviewed. Third, the role of people–place bonding in the promotion of refugee well-being is discussed.

### 3.1. Mental Illness and Well-Being in Refugees

Refugees are one of the most vulnerable groups in our society [3]. They have had to leave their countries, homes, social networks and work due to life-threatening circumstances, such as the fear of being persecuted or threatened by other people and/or environmental hazards. Many refugees go through a perilous journey in search of safety, leaving many traumatized and suffering from mental health problems [4]. A systematic review estimated the chance that refugees in Western countries suffer from post-traumatic stress disorder (PTSD) as ten times more likely than that of the local age-matched population. They also found that the prevalence of serious mental disorders was as high as 9% for PTSD and 5% for major depression, with high co-morbidity [5]. Refugees also have a high risk for suicide and social exclusion. Due to these risk factors, there is a large need for interventions that improve refugee mental health [4].

Post-migration experiences have even more impact in terms of undermining refugees’ well-being than pre-migration experiences or trauma [6]. Refugees have to rebuild their lives, often in an environment where they do not feel welcome [7]. During resettlement, refugees are continuously exposed to stressors, such as social, economic and social insecurity, which negatively affect their psychological functioning [8]. Refugees are therefore at great risk of developing mental health problems during the time of resettlement. Poor social integration partly explains the high rates of long-term mental disorders [9] observed in resettled refugees.

To obtain a complete picture of refugee’s mental health, in addition to mental illness, positive mental health also has to be considered. Positive mental health, or well-being, is more than the absence of ill-being. Well-being and psychopathology are two related but distinct dimensions of mental health [10]. Well-being, which is studied in the field of positive psychology [11], is beneficial for several outcomes such as coping during difficult times, problem solving, performance in complex mental tasks, and various health outcomes [12].

There are three dimensions of well-being. First, emotional or *subjective* well-being refers to experiencing positive emotions in relation to negative emotions and satisfaction with life [13]. Second, *psychological* well-being is defined as positive growth, environmental mastery, and positive relationships with others and self-acceptance [14]. Third, *social* well-being is about social functioning and encompasses the dimensions of social coherence, acceptance, actualization, integration and actualization [15]. In order to flourish, all three of these types of well-being are important [10].

When examining determinants of well-being for refugees, coping with adversity is especially important. *Resilience* (the ability to bounce back after adversity) plays an important role in well-being [16]. On a personal level, self-esteem and positive adaptability to stress are key factors to developing resilience. On a social level, support from families, family cohesion, and peer support facilitate resilience. According to Fielding and Anderson [17], collective resilience facilitates resettlement of refugees by building communities that support refugees in recovering from trauma. It has been shown that resilience can be improved with specific interventions [18].

Traumatic experiences can also lead to *post-traumatic growth* [19]. It can develop as a result of the cognitive processes that are initiated in order to cope with traumatic events, leading to thriving and personal growth post-trauma, resulting in new self-insights, a new sense of meaning and purpose in life, changes in perception, relationships and life priorities [16]. In order to develop post-traumatic growth, *hope* is vital. A hopeful disposition despite challenging circumstances is a protective factor and helps refugees to acculturate and feel empowered during resettlement in the post-migration phase [20].

Several meta-analyses indicate that positive psychology interventions (PPIs) can enhance well-being in the general population and in vulnerable groups, such as those with psychiatric disorders [21,22,23]. Improving well-being is another approach to protecting people from developing mental disorders and increasing the likelihood of recovering from mental illness [24,25,26]. Therefore, addressing well-being in refugees is a promising approach [27], especially because, in many cultures, receiving help for psychological problems is stigmatized and therefore, might be rejected by refugees. Still, PPIs for refugees are rare. One PPI developed for refugee children in Greek refugee camps showed improvements in their well-being, self-esteem, and optimism and depressive symptoms, with the children especially valuing building their strengths and the new sense of togetherness [28].

In conclusion, in order to obtain well-being, integration into society is vital, for example via work. While European countries seek ways to integrate refugees into society, little is known on how to optimally help this vulnerable group. There is a lack of both a theoretical framework and consequently a lack of evaluated refugee mental health interventions. Yet, in order to be able to find viable solutions to promote well-being of refugees and their integration into society, we need such a theoretical framework. Only when we can base our practical efforts on sound theory can we develop interventions, projects, and initiatives that will work optimally which can be further improved in a systematic way in the future.

### 3.2. Self-Determination Theory: A Helpful Framework for Positive Integration

A framework explaining the needs and determinants for refugee well-being is Self-Determination Theory (SDT) [29]. SDT is a well-researched theory of motivation and offers a theoretical explanation for how to improve well-being. SDT postulates that every human being has three basic psychological needs: autonomy (the feeling of being the director of one’s own life), competence (the feeling of being good at something) and relatedness (the feeling of being connected with others). All three are required in order to reach and maintain a good level of well-being [30].

These basic needs are likely to be unsatisfied in the lives of many refugees, as they have lost control of their lives, are unable to use their skills in a job or hobby, and have lost their social networks. Integration into society fulfills the needs of refugees. For example, integration into a social group, such as one’s neighborhood or a religious community, will create new social bonds and can thus fulfill the need for relatedness. Being an active member of a club, for example, the local football club or a choir, can promote relatedness via shared interests. Moreover, participating actively in a hobby can support the feeling of being good at something, or of having the capacity of becoming better at a skill, and thus fulfills the need for competence. Organizations that support integration should leave decisions up to the refugee to support his or her autonomy.

*Work* is another highly successful way to achieve integration. Finding employment can satisfy basic psychological needs as it allows one to regain autonomy, use one’s competencies, and feel connected to colleagues at work [31]. Finding a job may not only help refugees to provide for themselves but could also improve their well-being and lay the foundation for integration. As an approach to helping people to fulfill their needs, interventions, including job-seeker programs that support refugees in findings employment, should focus on improving competencies (specific to the job they want to apply for), fostering autonomy (regaining a feeling of being in control by actively searching for a job) and relatedness (by building a support network).

### 3.3. Place Attachment: A Person–Place Bond in the Context of Resettlement

Satisfaction of the basic psychological needs and well-being can also be improved by *place attachment*. Visualization of a meaningful place, a place where one feels emotionally connected to, has been found to improve self-esteem, belonging, and meaning, including for people who experience need threats [32]. Place attachment is thus a relevant concept for understanding refugees’ resettlement experience, for whom the satisfaction of their needs is not a given.

Place attachment has been defined as a multidimensional affective bond between people and places, involving a symbolic relationship with the place and the willingness to maintaining proximity with the place [33,34,35,36]. The traditional conceptualization of place attachment focuses on residential stability and the creation of person–place bonds through time, habits, and place-related life experiences [37,38]. However, this conceptualization is thought to be simplistic, since place attachment can be ambivalent [39], constitutes various types (traditional and active), and is non-attached (alienation, place relativity, and placelessness) [40]. A growing body of research in recent years has focused on how people develop affective bonds with new places. This includes a focus on contemporary mobility patterns that are characterized not only by migrations but also by a general reduced residential stability [40,41,42].

### 3.4. Developing a Bond with a New Place: The Role of Self-Continuity and of Need Satisfaction

Most of the literature on bonding to places of resettlement has focused on the roles of self-continuity and place making. The concept of self-continuity was first introduced by Hallowell [43] and later incorporated into the identity process theory by Breakwell [44] as the need for the self to be organized through a coherent “story” that links past and present behaviors. Based on Breakwell’s model, Twigger-Ross and Uzzell [45] include self-continuity among the functions of the person–place relationship. For instance, people are known to bond more easily to places with a climate similar to the place they come from [46].

Consistently, several studies conducted by Shampa Mazumdar, Sanjoy Mazumdar and colleagues show that self-continuity is a relevant factor to create a bond with new places of settlement for migrants. In a study on the Vietnamese enclave of Little Saigon in Westminster, California, the architectural environment and the immigrants’ social, commercial, and ritual activities were found to contribute to the sense of place of the local Vietnamese community, thus enabling it to simultaneously remain connected to the places of origin while developing significant new place ties [47].

The relevance of continuity is also highlighted in within-country migration processes, usually from rural to urban areas. For instance, Becerra, Merino, Webb, and Larrañaga [48] showed how indigenous Mapuche, moving from rural areas to the megalopolis of Santiago de Chile, strengthen their relationship with the new place by recreating and locally translating traditional Mapuche practices. Similar studies have been conducted in China, by Liu, Fu, Van den Bosch, and colleagues [49], showing that integrating landscape elements which are familiar to newcomers could ease the development of place attachment to new urban places of residence. The importance of the role of familiar elements in easing bonding to new places has also been shown by Cheng and Kuo [50] with an experimental design and participants from Macao and Taiwan.

Continuity is often achieved through place making, i.e., the possibility of migrants changing private and public spaces to make them look more familiar and similar to places from their country of origin. Place making can aim to make the place look more familiar in terms of design, materials, and practices, with a particular emphasis on the creation of spaces for cultural and religious practices, via place planning and organization, place design, and place rituals [51,52,53,54].

Public green areas, such as parks, can help recall the rural landscapes of the home country and thus can be a privileged setting for place bonding, place making and cultural practices among migrants, as shown in studies conducted both in Chile [55,56] and in the USA [57]. Place-making can also be the result of participation in local initiatives, such as gardening, an activity that is practiced by many migrants coming from agrarian backgrounds. This can take place within a diverse community, promoting interactions between migrants and locals, with material and emotional benefits for the newcomers [58,59] as well as the locals.

Although most of the literature on place attachment and newcomers focuses on the role of self-continuity, a few very innovative studies focus on the role of basic need satisfaction in the development of place attachment, giving a new perspective that can be directly linked to well-being.

### 3.5. A New Perspective: Developing Place Attachment as the Result of Basic Need Satisfaction

Even though need satisfaction has mainly been examined in different contexts such as well-being and secure interpersonal attachment (e.g., La Guardia et al. [60]), some new studies have started to show the role of need satisfaction for the development of place attachment (e.g., Van Riper, Yoon, Kyle, Wallen, Landon, and Raymond [61]). Landon et al. [62] found a correlation between place attachment and basic need satisfaction (as proposed by SDT [60]). Among visitors of wild areas in the Southern Appalachian Region, perceiving a landscape as supporting autonomy, relatedness, and competence was associated with identification, dependence, and emotional connection with that landscape [62]. Similarly, a recent study on place attachment in the context of environmental risk showed that manipulating an evacuation site to show it as more fit to satisfy individuals’ basic psychological needs made participants more likely to declare greater attachment towards it, and more likely to evacuate in case of environmental emergency [63]. These findings do suggest that people develop affective bonds with places that satisfy one or more of their basic needs. It would not be unusual then to observe that it is likely that refugees are able to develop an affective bond with places that satisfy their basic psychological needs.

Other results, even if not directly referencing SDT and basic need satisfaction, suggest a link between need satisfaction and the development of a person–place bond [32,64]. Some studies on the topic of place bonding in the context of resettlement also give some hints in this direction. Sampson and Gifford [65], for instance, in a study on young refugees in Australia, found that four kinds of places (places of opportunity, places of restoration, places of sociability, and places of safety) are particularly important to young refugees, as they have a therapeutic and restorative role for newcomers and contribute to their well-being. Among these places, places of opportunities (often schools, but also places for leisure time such as parks) seem to be able to support autonomy and competence, allowing youths to conduct the activities they want to feel competent at. Places of sociability instead refer to the importance of relatedness. Places of safety and places of restoration, however, make reference to needs—i.e., restorativeness, relaxation, safety—that are not part of SDT, but that are commonly known in the place attachment literature.

Similarly, a study on refugee women from Myanmar in New Zealand that was part of a multisensory research project found that refugee participation in local initiatives and familiarization with local places of need and of pleasure reduced their stress and anxiety. Additionally, it increased their feelings of safety, autonomy and belonging, ultimately contributing to building place attachment [66]. The relevance of relatedness is reported by many studies, such as a study about Ukrainian immigrants in Poland, where it was found that migrants who had strong ties with the Polish people were more strongly attached to Warsaw, which in turn increased their willingness to stay in Poland [67]. A Dutch study on Syrian refugees reported that many of those who had been assigned a home in small Dutch communities often moved to bigger urban areas where more Syrians could be found [68]. Finally, a study on Afghan refugees in Finland highlighted how different individuals from the same community have different resettlement experiences and need different adapters to establish a successful relationship with the new place [69]. This suggests that the UN Refugee Agency, UNHCR, and institutions working in refugees resettlement processes should take these different place attachment strategies into account for easing refugees’ well-being and successful resettlement [69]. More broadly, in the general population, correlation studies show that carrying out personally significant and involving activities in a given place, is associated with developing and strengthening the psychological meaning of that place for the actor (e.g., in terms of place identity [70]).

These few examples seem to draw a connecting line between need satisfaction and place attachment, often by means of specifically directed social-psychologically meaningful activities, thus making a stronger and more relevant link among the environment, the person’s place attachment and their resulting well-being. This highlights the importance of successful place bonding for refugees’ well-being. However, more research is needed to investigate how to integrate the different approaches and results. Many questions remain to be answered such as what is the role of self-continuity in a need satisfaction framework? Is it just another need? How many other needs are there as well as the well-known basic psychological ones (see Ariccio, Lema-Blanco and Bonaiuto) [63]?

### 3.6. Developing Place Attachment towards Places of Resettlement: A Dynamic Process

The process of creating a successful person–place bond in the resettlement location is complex, varied, and dynamic. Some of the literature has recently focused on this last feature, i.e., on what necessary steps or phases help to develop a successful person–place bond in a new place. Even though there is an agreement that this process goes through several steps, no consensus has yet been reached on which steps these are. This is also consistent with the ongoing debate about the geometry and mutual relationships between the different person–place constructs [42,71,72]. For instance, the concept of place identity is considered as the cognitive facet of the person–place relationship, contributing therefore together with personal and social identity to the definition of the individual’s identity. Knez [46] showed in a study about residents’ relationship with the Swedish town of Gothenburg that five dimensions of place identity (place-related distinctiveness, place-referent continuity, place-congruent continuity, place-related self-esteem, place-related self-efficacy) are all predicted by place attachment which in itself is predicted by length of residence.

Two studies on place attachment and place identity in natives and non-natives conducted in Spain suggest that place attachment develops before place identity, at least in the case of the non-natives [73]. A contrasting result is the one proposed by Kyle, Jun, and Absher [74] in a study about residents living in the wildland–urban interface outside of San Diego and Los Angeles. They found that the tripartite organization of human–place bonding includes different steps, with place identity working as an antecedent to place attachment and place dependence. This is consistent with the identity theory [75].

Similarly, Trąbka [76] conducted a qualitative study on Polish migrants living in London and Oslo and found four phases of person–place bonding in the migration context: place dependence, place discovered, place identity, and place inherited. Place dependence is the functional bond to place that accounts for the relationship people establish with a place due to the activities or tasks that the place enables (see also [77]). Place discovered is defined by Trąbka [76] as encompassing “both behavioral and emotional aspect, and it is best observed among those participants who gain enjoyment, aesthetic pleasure and a sense of mastery from an intentional familiarizing with the city as such” (p. 70). Place identity is considered as a cognitive effect of place dependence that makes the place become part of the self. Place inherited is defined as “a strong and taken for granted bond, prevalent among people who have spent their whole life in one place, who have strong family connections in it and who cannot imagine leaving” [76] (p. 71). According to Trąbka [76], it is strongly associated with length of residence and it can be found in refugees who have been in London or Oslo for more than ten years and often arrived there at a young age. These phases of person–place bonding often coexist and may emerge gradually in the process of adaptation to a new place, contributing to the overarching dimension of place attachment, with a person–place relationship that evolves from functional bonds to feelings of belonging. Another study about people living abroad for long-term periods in a variety of countries highlights the coexistence of different kinds of person–place relationships between the individual and the host country [78]. Individuals seem to have multi-local identities, feeling like tourists, immigrants, and locals in their new settlement. Even though a temporal continuum can be found in the emergence of these identities, each can coexist within the same individual, triggered by everyday micro-moments and by the relationship with their countries of origin and their country of resettlement [78].

## 4. Discussion

This paper presented a literature review on the role of place attachment for refugee well-being, focusing particularly on how this bond develops both in term of antecedents and as a dynamic process. The literature on this topic is fragmented. We focused primarily on refugees, who come from contexts of war and lack of safety and are thus particularly fragile from a psychological point of view. This makes it especially relevant to investigate how their well-being can be improved in resettlement contexts. We also discussed studies on people that do not strictly fit into the definition of “refugees” (i.e., migrants, intra-state migrants), as it is plausible that the psychological processes of bonding to a new place are similar for non-refugees that move to a different country. It should also be noted that when dealing with people who move to a different country for a long-term stay, different terminologies are employed depending on the legal status of these individuals and their backgrounds and motivations for travelling, e.g., “refugee”, “migrant” or “asylum seeker” [79], although their resettlement experiences may be similar.

The literature summarized in this paper shows that a promising approach to refugee resettlement is focusing on positive aspects, such as hope and support of need fulfillment to support refugees instead of the traditional focus on (solving) problems. SDT is a practical approach to helping refugees (who often suffer from mental health problems) by promoting their positive mental health. Supporting autonomy, relatedness, and competence can be attempted in various ways that can be adapted to each specific context. Professionals working with refugees, who are often used to a problem-based approach, would likely need additional training to implement this very different approach.

For well-being interventions to be effective, it is fundamental that all three basic psychological needs are addressed. For example, merely focusing on fostering competence development (e.g., interventions aiming at developing job applying skills) will not be as effective on refugees’ well-being if autonomy and relatedness are not also taken into account. In general, interventions should be based on the actual needs of refugees. Refugees should be involved in the development of interventions as much as possible. This will increase their sense of autonomy and relatedness in their integration process. Unfortunately, few evaluated well-being interventions for refugees are available that have an emphasis on people–place bonding processes.

Migration processes are one of the contemporary phenomena that question traditional environmental psychology constructs such as the idea of a singular, meaningful place for the individual. Research is growing on how people create new person–place bonds while maintaining the old ones and simultaneously adapting to changing places. This highlights the role of continuity, empowerment, and place making as ways to maintain and create new bonds (e.g., Di Masso et al. [42]). However, studies on migration show that bonding with a place can be a long and articulated process, whose different steps and dynamics are still to be defined and investigated [41,76]. In this sense, migration and refugees are contemporary social phenomenon that provide an interesting angle on the constructs and theories usually studied by person–place relationship studies.

On the other hand, understanding more of how refugees develop a bond with their new places of settlement would be of fundamental importance to ease refugees’ integration and promote their well-being. For instance, it would be fruitful to allow refugees to bond with a place that has familiarity with their original place of attachment. This is consistent with the importance of continuity. In this sense, enabling place-making would also be an efficient strategy, since refugees would be able to actively improve the place, adding familiarity and thus easing the bonding process. Similarly, consistent with SDT, it should be possible to facilitate person–place bonds in a migration context by making places of settlement able to satisfy refugees’ psychological basic needs for autonomy, competence and connectedness.

The role of place attachment within the SDT framework is especially promising for the role of people–place bonds in the process of well-being promotion. The development of place attachment seems to be affected by self-continuity and by the satisfaction of psychological needs. However, it is not clear how these are inter-related. Is self-continuity a need specific to place attachment in relocation contexts? Are there other needs to be satisfied (e.g., safety, relaxation)? Several studies on environmental psychology try to list the needs places can satisfy and their role in the development of place-bonding [32,63,64]. As there is still a lack of systematization of these needs and especially of integration with existing theories such as SDT, more research is needed in order to optimize the effect of well-being interventions for refugees.

## 5. Conclusions

Refugees are a vulnerable group at risk for mental health problems and lesser well-being, with their potential often being undervalued or neglected in their new resettlement environment. To understand their situation and to support positive integration, Self-Determination Theory is a useful framework. If the need for autonomy, competence and relatedness is supported, the circumstances for positive mental health are shaped. Little is still known about the steps involved in place bonding which is consistent with the ongoing debate about the geometry of the psychological constructs of person–place relationship. Yet, knowing more about needs to be satisfied for easing place bonding could be of capital importance for facilitating refugee well-being. Ultimately, improving knowledge and understandings of the phases of this dynamic process could be useful for a more successful implementation of refugee resettlement practices and activities.

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
