# Peer review of "The Role of Place Attachment in Promoting Refugees’ Well-Being and Resettlement: A Literature Review"

_ijerph, 2021, doi:10.3390/ijerph182111021_

Round 1

Reviewer 1 Report

Dear Authors, thank you for submitting the article to the IJERPH. The subject matter of the article fits the scope of the journal, however much work is needed to approve the article and make it publishable.

Here my main concerns:

  1. What is the research aim/objective/research question you address in the article? (The abstract is entirely descriptive as to the case).
  2. Also a literature review has a research question - why is it important to review the literature? What is the contribution of your review above and beyond the literature that exists and is reviewed in your article?
  3. What methodology did you use? Have a look at articles which deal with literature reviews, there are methodologies to it!
  4. What criteria did you use to chose articles reviewed? Why those and not others? Present your method and data.
  5. What can we learn from your literature review? Contribution? 

I hope these comments will help you to improve the article substantially.

Reviewer 2 Report

This is a very interesting conceptual paper and  will be of value to researchers and practitioners working in the area of refugee resettlement.

This is a literature review and thus should be introduced as such. It should be clearly stated in the Abstract that this is a literature review as stated by the authors  on page 7 (Discussion) of the manuscript and its overall purpose of discussing different aspects of determining refugee wellbeing:

"This paper presents a literature review on the role of place attachment for refugees’ well-being, focusing particularly on how this bond develops both in term of antecedents and in term of dynamic steps". 

To provide greater clarity to the authors points,  editing, wording/word choice,  and phrasing throughout the manuscript should be reviewed. For instance on page 2, suggest stating the "dimensions" of wellbeing instead of  "types" of wellbeing.  On page 3, paragraph 1, line 6, instead of saying "culture of origin", suggest using "in many cultures"  and page 3, paragraph 3, line 1  instead of saying, "A framework that can explain needs and determinants  suggest... "A framework explaining needs and determinants..." 

The different sections of the paper need to be edited for clarity and succinctness and should be introduced in the introductory paragraph as the areas that will be reviewed or discussed related to refugee wellbeing.   Also, in the introduction, the authors should describe how they conducted their literature review and selected the studies that they included to make their points and advance their thesis about refugee wellbeing, and components attached to describe its importance.

Author Response

see attachement

Reviewer 3 Report

The article provides a good overview of the existing literature on place attachment, what is lacking are solutions for the refugees. The article is worthy of publication per se as it provides a handy overview of the state of the art, nevertheless the suggestion to improve it is to adding at least some persperspectives on how future research on the field should be conducted (e.g. what it should address, toward which solutions).

Round 2

Reviewer 1 Report

Dear authors, thanks for the resubmission. You substantially improved the article which is now ready for publication. 

Good luck with your further research. 

Author Response

Dear reviewer, 

Thank you kindly for your time and effort in helping us improve the quality of our paper. 

We wish you all the best, 

Thomas Albers 

Reviewer 2 Report

This is an important topic and the authors have made a gallant effort to revise the manuscript.  As one of the major issue observed is wording and clarity,  attached is suggested revisions. Hopefully, they are not a departure from the authors intent and focus of the paper, will be helpful to the authors.

Additional suggestions are:

  1. Be sure to clearly state the purpose of the paper
  2. In reporting the results of the literature search, please specify all the search engines used, how many articles/papers met the selection criteria, and what specific time period the search and review covered-was it 10 years? 15 years?
  3.  The authors mentioned they expanded their search to include papers related to migrants in addition to refugees. Can the authors report how many or what percentage of the papers were on migrants? 
  4. If overall, the numbers of studies meeting criteria were small, then a chart or table listing each study, authors, focus of paper would be helpful to include in the paper so that it can be clearly seen which papers were used for the review. Another approach may be to identify those studies in the reference list with an asterisk (*)

Author Response

Pls see attached
